# Exploiting Dynamic Sparsity in Einsum

**Christoph Staudt**\*, **Mark Blacher, Tim Hoffmann,**
**Kaspar Kasche, Olaf Beyersdorff, Joachim Giesen**
Friedrich Schiller University Jena

## Abstract

Einsum expressions specify an output tensor in terms of several input tensors. They offer a simple yet expressive abstraction for many computational tasks in artificial intelligence and beyond. However, evaluating einsum expressions poses hard algorithmic problems that depend on the representation of the tensors. Two popular representations are multidimensional arrays and coordinate lists. The latter is a more compact representation for sparse tensors, that is, tensors where a significant proportion of the entries are zero. So far, however, most of the popular einsum implementations use the multidimensional array representation for tensors. Here, we show on a non-trivial example that, when evaluating einsum expressions, coordinate lists can be exponentially more efficient than multidimensional arrays. In practice, however, coordinate lists can also be significantly less efficient than multidimensional arrays, but it is hard to decide from the input tensors whether this will be the case. Sparsity evolves dynamically in intermediate tensors during the evaluation of an einsum expression. Therefore, we introduce a hybrid solution where the representation is switched on the fly from multidimensional arrays to coordinate lists depending on the sparsity of the remaining tensors. In our experiments on established benchmark einsum expressions, the hybrid solution is consistently competitive with or outperforms the better of the two static representations.

## 1   Introduction

Einsum was introduced in NumPy [19] in 2011 and has since become the quasi-standard for specifying pure tensor expressions, meaning expressions that involve no non-linear operations on tensor entries. It is now an integral part of machine learning frameworks such as TensorFlow [1], PyTorch [25], and Nvidia's cuQuantum [5] library for quantum information science. So far, these frameworks support only dense tensor formats in einsum. While dense formats are widely used, sparse tensor formats such as coordinate lists offer potential advantages when many tensor entries are zero. They can reduce the number of floating-point operations (flops) compared to dense formats and, as we demonstrate in this paper, may even yield exponential improvements in specific cases. However, sparse formats also have drawbacks. They may require more memory when tensors are not particularly sparse, and while in theory their space usage grows at most linearly, in practice the memory overhead can be considerable. Moreover, they often use a less favorable data layout for computation, which can lead to significantly slower execution. Thus, choosing the appropriate tensor format is critical for efficient execution of einsum expressions.

Sparse tensor formats in einsum are supported in the Python package *sparse* [26] and also in SQL [8]. These implementations, however, rely on static format choices. In our experiments, we show that sparsity can evolve dynamically in intermediate tensors during the evaluation of an einsum expression, especially for tensor expressions that involve a large number of input tensors. Such expressions

---

\*Corresponding author: `christoph.staudt@uni-jena.de`

39th Conference on Neural Information Processing Systems (NeurIPS 2025).

are not yet a primary focus of established frameworks, but are becoming increasingly important in model classes such as probabilistic circuits [15, 28, 32] and probabilistic neural circuits [38], which often rely on einsum as a computational backend, as well as in neuro-symbolic models that combine deep learning with logical constraints [10, 12]. Moreover, the simulation and validation of quantum machine learning algorithms [7, 4, 35, 29] also benefit directly from dynamically sparse einsum. However, since sparsity can evolve dynamically, it is difficult to determine from the expression and the input tensors alone which tensor format is best for a given einsum expression. Therefore, to efficiently exploit dynamic sparsity in einsum, we introduce a hybrid solution, where the representation is switched on the fly from a dense to a sparse format once the sparsity of the remaining tensors falls below a predefined threshold. In our experiments we show that the overhead for switching from one format to the other is relatively low. As a result, the hybrid solution is consistently competitive with or outperforms the better of the two static representations.

Our approach to einsum aligns well with a current trend in the machine learning community that is shifting from traditional static sparse modeling to dynamic sparsity [17, 21, 24, 31, 33, 34, 37], which is broadly defined as any kind of sparse computation or memory compression where the sparsity pattern is input-dependent or adaptive [23].

## 2 Einsum expressions and tensor formats

In this section, we provide the necessary background on einsum expressions and their evaluation, as well as dense and sparse formats for storing tensors. Einsum expressions are of the form $\text{einsum}\left(I_1, \ldots, I_n \to O; \ T_1, \ldots, T_n\right)$, where the $T_i$ are input tensors with index strings $I_i$, and $O$ is the index string of the output tensor. The index strings consist of symbols that represent tensor axes, sometimes also called dimensions or modes. The size of the index string $I_i$ is the order of the tensor $T_i$. Notably, input tensors can themselves be einsum expressions.

### 2.1 Evaluating einsum expressions

All indices that appear in at least one input tensor but not in the output tensor are summed over. Indices that occur in two or more input tensors signify a combination, and if such indices are also summed over, they define a contraction. The simplest combination is the elementwise product of two vectors $u$ and $v$, written as $\text{einsum}\left(i, i \to i; \ u, v\right)$. The simplest contraction is the inner product $u^\top v = \text{einsum}\left(i, i \to; \ u, v\right)$, where the output index string is empty because the result is a scalar. More complex expressions involve multiple contraction indices, such as the matrix-matrix-vector product $\text{einsum}\left(ij, jk, k \to i; \ A, B, v\right)$, which results in a vector. Contracting over index $j$ first, as in $\text{einsum}\left(ik, k \to i; \ \text{einsum}\left(ij, jk \to ik; \ A, B\right), v\right)$, requires $2 \cdot |i| \cdot |j| \cdot |k| + 2 \cdot |i| \cdot |k|$ floating point operations (flops), whereas contracting over $k$ first, as in $\text{einsum}\left(ij, j \to i; \ A, \text{einsum}\left(jk, k \to j; \ B, v\right)\right)$, requires only $2 \cdot |j| \cdot |k| + 2 \cdot |i| \cdot |k|$ flops. Thus, the second contraction order is more efficient.

Computing an optimal contraction order that minimizes the number of flops is, in general, an NP-hard problem [11]. Therefore, heuristics such as greedy strategies [13] or graph partitioning methods [18, 30] are used in practical solvers for computing tensor contraction orders. The number of flops depends not only on the contraction order but also on the choice of data structure for representing tensors. Sparse tensor formats can significantly reduce the number of flops compared to dense formats when the tensors involved in the contractions are sparse. Note, however, that in einsum expressions involving a large number of tensors, it is not known in advance whether intermediate tensors will become sparse.

### 2.2 Dense and sparse tensor formats

In machine learning contexts, tensors are often equated with multidimensional arrays. Here, we take a more abstract view and consider tensors as multivariate functions of the form $T : A \to \mathbb{R}$, where $A = A_1 \times \ldots \times A_n$ with finite sets $A_i$, which are called tensor axes. The natural data structure for storing a tensor $T$ is a multidimensional array, where $T[a_1, \ldots, a_n] = T(a_1, \ldots, a_n)$. Alternatively, tensors can be stored as lists of coordinate entries of the form $\left(a_1, \ldots, a_n, T(a_1, \ldots, a_n)\right)$. Lists have the advantage that sparse tensors, meaning tensors in which a significant proportion of function values are zero, can be stored more compactly by only storing the coordinates where the function

values are nonzero. For dense tensors, however, the list format is less efficient because of the overhead incurred by explicitly storing the coordinates.

# 3 Exponential separation

In this section, we show that there exist einsum expressions that can be evaluated exponentially more efficiently when using coordinate lists instead of multidimensional arrays. These expressions are represented as tensor hypernetworks, which form the underlying algebraic structure of einsum expressions. The tensor hypernetworks used to demonstrate the exponential separation between data structures stem from model counting problems. The model counting problem #SAT is of independent interest in artificial intelligence, because a variety of real-world questions can be encoded very naturally and succinctly in #SAT, including problems in probabilistic reasoning [2, 20], risk analysis [14, 36], and explainable artificial intelligence [3, 27].

## 3.1 Tensor hypernetworks and einsum

Tensor hypernetworks define an output tensor in terms of input tensors. Formally, let $\mathcal{A} = \{A_1, \ldots, A_m\}$ be a set of axes. A tensor hypernetwork $\mathcal{T}$ is given by a finite set $\{T_1, \ldots, T_n\}$ of input tensors such that the axes of each input tensor $T_i$ are a subset $\mathcal{A}_i \subseteq \mathcal{A}$ of the axes set. The subset $\mathcal{A}_i$ is encoded in the index string $I_i$ of the corresponding einsum expression. For instance, the tensor hypernetwork corresponding to the einsum expression einsum $(ij, jk, k \to i;\ A, B, v)$ from the previous section has three axes. The first axis, $i$, is used only by the matrix $A$. The second axis, $j$, is shared by the matrices $A$ and $B$. The third axis, $k$, is shared by the matrix $B$ and the vector $v$. Therefore, the set of axes for the matrix $A$ contains the first and second axes, indexed by $ij$. The set of axes of the matrix $B$ contains the second and third axes, indexed by $jk$, and the set of axes of the vector $v$ contains the third axis, indexed by $k$.

The output tensor is given by combining input tensors that share a common axis and by summing over all axes in $\mathcal{A}$ that are not part of the output tensor. Here, in order to keep the exposition simple, we assume that the output tensor has no axes, that is, it is a scalar, and thus we sum over all axes in $\mathcal{A}$. The *full combination* of the tensor hypernetwork $\mathcal{T}$ is a tensor $T$ with axes $\mathcal{A}$, such that, for $a \in A$, $T(a) = \prod_{i=1}^{n} T_i(a_{|_i})$, where $a_{|_i}$ is the projection of $a$ onto the axes in $\mathcal{A}_i$. The *full summation* of $\mathcal{T}$, that is, the scalar output tensor, is the sum of all function values of $T$ and can thus be written as

$$\sum_{a \in A} T(a) = \sum_{a \in A} \prod_{i=1}^{n} T_i(a_{|_i}).$$

Algorithmically, of course, it is not a good idea to first compute the full combination and then the full contraction, because the full combination results in a large intermediate tensor. In practice, the tensor hypernetwork is contracted by contracting over the axes one after the other, following an optimized contraction order. To contract over an axis, one collects all tensors that share the axis, computes the full combination of these tensors, and sums it over the given axis. Finally, the collected tensors are replaced by the sum over the full combination, which is a single tensor but not necessarily a scalar.

The combinatorial structure of a tensor hypernetwork can be encoded in a hypergraph: the nodes are given by the input tensors in $\mathcal{T}$ and for each axis $A_i$ there is a hyperedge that contains all input tensors $T_j$ with $A_i \in \mathcal{A}_j$. Contracting over an axis in a tensor hypernetwork corresponds to the contraction of the corresponding hyperedge in the hypergraph. The contraction of a hyperedge is illustrated by example in Figure 1.

## 3.2 Model counting by tensor contractions

Problems on propositional formulas naturally fit into the framework of tensor hypernetworks [6, 16]. A propositional formula is a multivariate function of the form $F : \{0, 1\}^n \to \{0, 1\}$ and thus a tensor with axes $A_i = \{0, 1\}$ for $i \in [n]$.

Given a propositional formula $F$, the satisfiability problem SAT asks if there exists $a \in \{0, 1\}^n$ such that $F(a) = 1$. Such a satisfying assignment is called a *model* for $F$. The model counting problem #SAT asks to count all models, that is, all satisfying assignments.

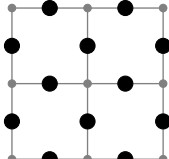 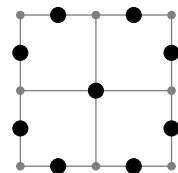

Figure 1: Left: Tensor hypernetwork for the grid formula $\text{GRID}_3$ (cf. Section 3.3). Here, large black bullets represent input tensors (matrices), and small gray bullets represent hyperedges. The tensor hypernetwork has one hyperedge of size four (at the center), and four hyperedges of sizes three and two, respectively. Right: The tensor hypernetwork after contracting the hyperedge of size four. Four second-order tensors have been contracted into one fourth-order tensor.

Both problems, satisfiability and model counting, can be served by tensor hypernetwork contractions. This is possible because any propositional formula has a representation in conjunctive normal form (CNF), that is, as a conjunction of clauses. A clause is a disjunction of literals, and a literal is either a variable $x_i$ or its negation $\overline{x}_i$.

Let $F$ be a propositional formula in conjunctive normal form, $\text{clauses}(F)$ be its clauses, and $\text{vars}(F)$ be the set of variables in $F$. For a clause $C \in \text{clauses}(F)$, let $\text{vars}(C) \subseteq \text{vars}(F)$ be the set of variables in $C$. Since any clause $C$ is itself a propositional formula, it is also a tensor $T_C$. The axes of $T_C$ correspond to its variables $\text{vars}(C)$. The tensor hypernetwork $\mathcal{F}$ that results from the CNF representation of the formula $F$ is given as $\mathcal{F} = \{T_C \mid C \in \text{clauses}(F)\}$. The full combination of $\mathcal{F}$ corresponds to the formula $F$, and the full summation

$$\sum_{a \in \{0,1\}^n} \prod_{C \in \text{clauses}(F)} T_C(a_{|C}),$$

where $a_{|C}$ is the projection of $a$ onto the axes of $T_C$ that correspond to $\text{vars}(C)$, is the propositional model count.

## 3.3 Exponential separation

We separate the two data structures, multidimensional arrays and coordinate lists, on a family of grid formulas. Let $[n] = \{1, \ldots, n\}$, the formula $\text{GRID}_n$ has variables $x_{i,j}$ for $i, j \in [n]$ and clauses

$$\neg x_{i,j} \vee x_{i+1,j} \text{ and } x_{i,j} \vee \neg x_{i+1,j}, \text{ for } i \in [n-1], j \in [n],$$
$$\neg x_{i,j} \vee x_{i,j+1} \text{ and } x_{i,j} \vee \neg x_{i,j+1}, \text{ for } i \in [n], j \in [n-1].$$

That is, $\text{GRID}_n$ has $4n(n-1)$ clauses and $n^2$ variables. The two clauses in the first line encode the condition $x_{i,j} = x_{i+1,j}$ and the two clauses in the second line encode $x_{i,j} = x_{i,j+1}$. That is, by construction, to satisfy $\text{GRID}_n$, all variables have to take the same value. Therefore, $\text{GRID}_n$ has exactly two models.

If the clauses are encoded in multidimensional arrays, then these multidimensional arrays are given by the following matrices $T_{\neg x_{i,j} \vee x_{i+1,j}} = \left(\begin{smallmatrix} 1 & 0 \\ 1 & 1 \end{smallmatrix}\right)$ and $T_{x_{i,j} \vee \neg x_{i+1,j}} = \left(\begin{smallmatrix} 1 & 1 \\ 0 & 1 \end{smallmatrix}\right)$, where, in both matrices, the columns are labeled by $x_{i,j} = 0, 1$ and the rows are labeled by $x_{i+1,j} = 0, 1$, starting in the top left corner. The encoding of the clauses $\neg x_{i,j} \vee x_{i,j+1}$ and $x_{i,j} \vee \neg x_{i,j+1}$ into matrices follows analogously.

Contractions are over the axes, that is, the variables. The contractions over $x_{i,j}$ and $x_{i+1,j}$, respectively, involve both tensors $T_{\neg x_{i,j} \vee x_{i+1,j}}$ and $T_{x_{i,j} \vee \neg x_{i+1,j}}$, which are contracted by an elementwise multiplication. Therefore, the tensors $T_{\neg x_{i,j} \vee x_{i+1,j}}$ and $T_{x_{i,j} \vee \neg x_{i+1,j}}$ can be replaced with their elementwise product in the tensor hypernetwork encoding, which reduces the number of tensors to $2n(n-1)$ and is standard practice in contraction algorithms. In the multidimensional array encoding, all the $2n(n-1)$ tensors are $(2 \times 2)$-matrices with 1 on the diagonal and 0 on the off-diagonal. Their coordinate list encoding is given as $((0,0,1),(1,1,1))$.

**Lemma 1.** *Contracting the tensor hypernetwork encoding of $\text{GRID}_n, n \geq 2$ needs $4n^2 - 4n - 1$ flops, when using the coordinate list encoding.*

*Proof.* As we have explained before, the tensor hypernetwork is contracted by contracting over the $n^2$ axes one after the other. Since all axes for the $\text{GRID}_n$ tensor hypernetwork have size two, the

arguments of all tensors, including the intermediate tensors, can be encoded by bitstrings. The lengths of the bitstrings are the orders of the corresponding tensors. For the proof, we can assume that the bits are ordered by the contraction order of the corresponding axes. Therefore, the first bit always corresponds to the contraction axis.

We show by induction that any algorithm for contracting the tensor hypernetwork encoding of $\text{GRID}_n$ keeps the following invariant: All tensors that arise during the contraction, except for the last tensor, have exactly two non-zero elements, namely, at the all-zeroes and the all-ones bitstrings, where they evaluate to $1$. The last tensor gives the value of the tensor hypernetwork, which is a scalar.

**Base case.** By construction, the invariant holds true for the input tensors that encode the clauses as shown above.

**Inductive step.** If the invariant holds true before a contraction, then it also must hold true after the contraction, because, as we show below, computing the full combination for the given axis and the subsequent summation over this axis both keep the invariant.

*Full combination:* Assume that $c$ tensors need to be contracted. The value of the full combination at every bitstring is given by the product of the values of the tensors that need to be contracted at the projection of the bitstring onto the axes of these tensors. In particular, we only multiply elements whose first bit is equal. By our induction hypothesis all to be contracted tensors only take non-zero values at the all-zeroes and the all-ones bitstrings. Therefore, only two products are non-zero, namely the products at the all-ones and the all-zeroes bitstrings.

*Summation:* We need to sum all entries that differ on the first bit, but are equal on all other bits. Since the first bit is always equal to the other bits, no summation occurs, until we reach the last contraction, where only one bit is remaining. The summation for the last contraction always boils down to the addition $1 + 1$. This results in the correct model count of $2$.

*Required flops.* As discussed above, the full combination of $c$ tensors needs $2(c-1)$ flops. Overall we need to combine $2n(n-1)$ tensors leading to $2(2n(n-1)-1) = 4n(n-1) - 2$ flops. As only the final summation needs a flop, the total number of flops is $4n^2 - 4n - 1$. $\qquad\square$

The proof shows that, when intermediate tensors become large in terms of the number of axes, they become hyper-sparse. Next, we show that the intermediate tensors, actually become large in terms of the number of axes. The exponential separation then follows, because all entries must be computed in the dense tensor encoding.

**Lemma 2.** *Contracting the tensor hypernetwork encoding of $\text{GRID}_n$ needs at least $2^n$ flops, when using multidimensional arrays.*

*Proof.* Mostly follows from Proposition 4.2 in Markov and Shi [22]. A full proof is included in the appendix. $\qquad\square$

We summarize Lemmas 1 and 2 in the following proposition.

**Proposition 1.** *Contracting the tensor hypernetwork encoding of $\text{GRID}_n$ needs exponentially more flops, when using multidimensional arrays instead of coordinate list encodings.*

Here, the data structures can be separated exponentially, because the intermediate tensors become hyper-sparse. In general, it is hard to tell from the input tensors if intermediate tensors will become sparse enough to warrant the sparse representation, which can also incur significant overhead over the dense encoding. Even in the grid formulas the input tensors have an average sparsity of $50\%$, that is, only half of the entries are zero, which, in general, is not considered sparse.

## 4  Experiments

We first verify Proposition 1 empirically by simulating the model counting instance $\text{GRID}_n$. Since this is a synthetic result, we also analyze dynamic sparsity on a recent benchmark of 168 einsum expressions [9] from areas such as probabilistic models, weighted model counting, tensorized language models, and quantum computing. Our findings show that hyper-sparse instances occur in practice but often become evident only during execution. Sparse formats are efficient in such

cases but introduce overhead on dense instances. To address this, we propose a hybrid approach that performs well across the benchmark.

All experiments were run five times and we report median runtimes. Experiments were conducted on a 64-core system (4 × Intel Xeon Gold 6130, 2.1 GHz) with 1.5 TB RAM. Code is available at `https://github.com/ti2-group/dynamic-sparsity-einsum`. We use PyTorch [25] for the dense baseline, as it has been shown to perform best on the einsum benchmark. Existing sparse einsum libraries cannot handle all benchmark cases due to index limitations, especially in expressions with many input tensors and large intermediate tensors, so we use our own prototype implementation. We exclude ten dense instances from the einsum benchmark due to resource constraints, leaving 158 instances. Optimized contraction orders for all instances are provided by the benchmark.

## 4.1 Exponential speedup on $\text{GRID}_n$

To confirm Proposition 1 empirically, we executed the einsum expressions for model counting tensor hypernetworks $\text{GRID}_n$ for $n \in [2, 80]$ using both dense and sparse implementations. As shown in Figure 2, the exponential separation between the two tensor formats is clearly visible. Moreover, the figure illustrates that, as predicted by theory, the runtime for the dense tensor format grows exponentially in $n$, while the runtime for the sparse format grows only quadratically.

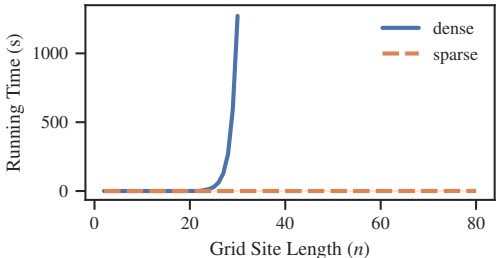
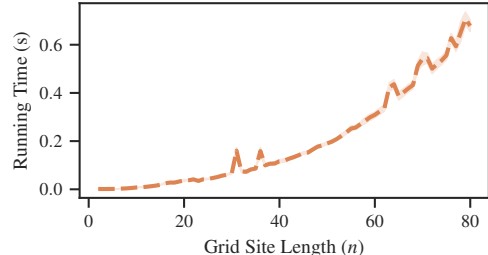

Figure 2: Running times for dense and sparse tensor formats when evaluating einsum expressions for $\text{GRID}_n$ formulas with $n \in [2, 80]$.

## 4.2 Dynamic sparsity in the benchmark

The results on the synthetic $\text{GRID}_n$ expressions show that the choice of tensor format can significantly impact the time needed to evaluate einsum expressions if tensors become sparse during the evaluation. We refer to the latter phenomenon as *dynamic sparsity*. To assess if dynamic sparsity also arises in practice, we analyzed the sparsity evolution, in particular the average density, during the evaluation of einsum expressions from the benchmark dataset.

**Average density.** In order to benefit from sparse tensor formats, the fraction of non-zero entries, and thus saved flops, needs to be small across the whole contraction sequence. Many sparse, but small contractions, will not lead to a significant speedup, if the running time is dominated by a few large and dense contractions. Therefore, we introduce $\text{avg}_\tau$ which denotes the *average density* of all tensors that arise in the contraction sequence up to the $\tau$-th contraction. Here, the average density is the sum of all non-zero entries in all tensors up to the $\tau$-th contraction, divided by the total number of elements across those tensors. The average density of the input tensors is $\text{avg}_0$. We refer to $\text{avg}_{\tau_{\max}}$ at the end of the contraction sequence simply as the average density of the expression.

**Density evolution.** We analyzed the average density of the benchmark expressions as a function of $\tau$. In 104 out of the 158 considered benchmark instances, the difference between the maximum and minimum average density exceeds 0.2, indicating significant changes in sparsity over the course of their contraction sequences. Figure 3 shows the evolution of average density for these instances. The instances are grouped by their final average density into three main categories. For each group, the bold red line represents the median average density evolution. A larger version of this figure is available in the appendix.

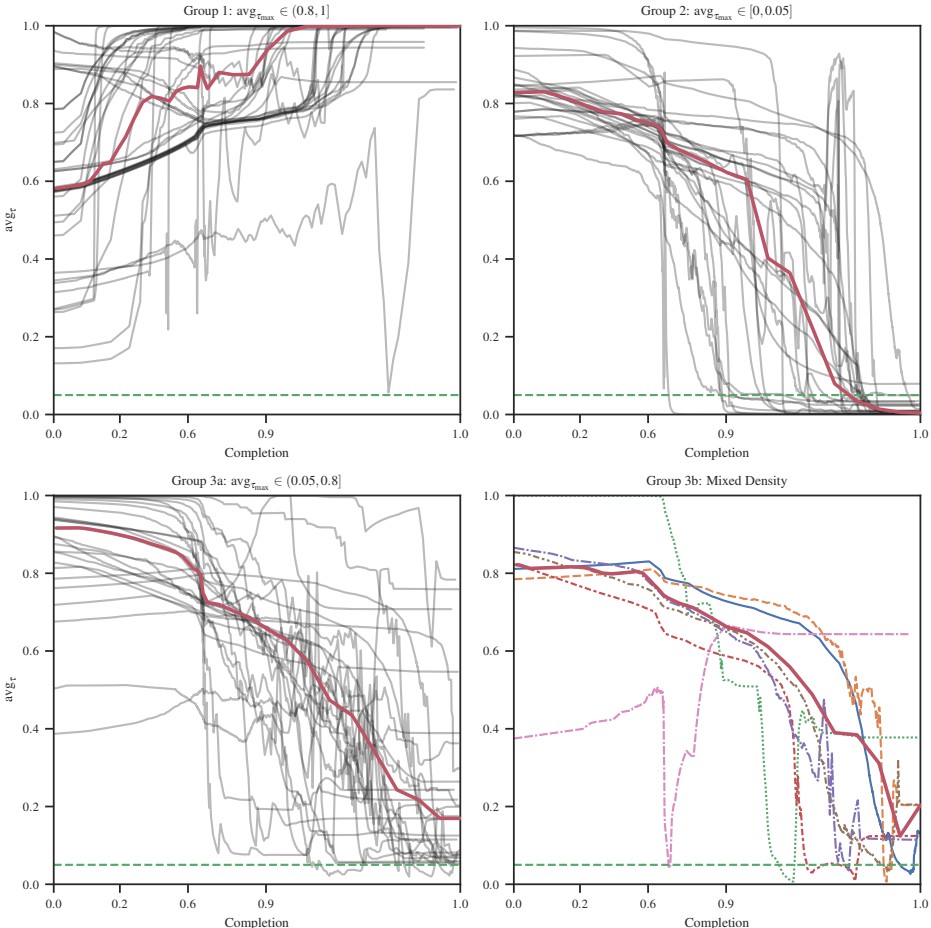

Figure 3: Evolution of average tensor density across instances, grouped by their final values. Within each group, a bold red line indicates the median density. The last group comprises instances transitioning from sparse ($< 5\%$) to dense ($> 10\%$). An exponential scaling at both ends is used along the x-axis to show the fraction of completed contractions $\tau / \tau_{\max}$. A green dashed line marks the 5% density threshold.

The first group contains 42 instances whose initial average densities $\text{avg}_0$ vary widely, but become nearly dense toward the end of the contraction sequence. Several instances begin relatively sparse, but then transition to dense tensors as zeros are eliminated through summations.

The second group contains 28 instances that become very sparse. Five of these instances have fewer than $10^{-5}$ non-zero entries. Interestingly, none of these instances starts out sparse. Their average sparsity arises only during large, flop-heavy contractions toward the end of the contraction sequence.

The third group contains 34 instances that gradually decrease in average density but maintain an average density above 5% and are split into two subgroups, Group 3a and Group 3b. Group 3b contains seven instances that exhibit more complex behavior. These instances first become sparse during the contraction sequence, but then their average density increases again. For instances that exhibit this mixed-density pattern, a sparse backend would not be optimal.

## 4.3 Sparse vs. dense on the benchmark dataset

The density evolution shows that there is potential for a sparse implementation to be faster than a dense implementation on at least 28 instances. We confirm this by executing all instances using both the sparse and dense tensor formats. Figure 4 shows the relative speedup, $(t_d/t_s)$, where $t_d$ is the running time for the dense format and $t_s$ is the running time for the sparse format. The red

line represents a speedup of one. Thus, points above the red line correspond to instances where the sparse format is faster than the dense format. This appears to be the case for the majority of instances with an average density below 5%. Afterwards, the dense format is mostly faster. In both cases, the speedup can reach up to 1,000x. Overall, the median slowdown of sparse vs. dense is around 17x on all instances and around 30x on dense instances. Thus, using the sparse format for all benchmark instances is not viable in practice.

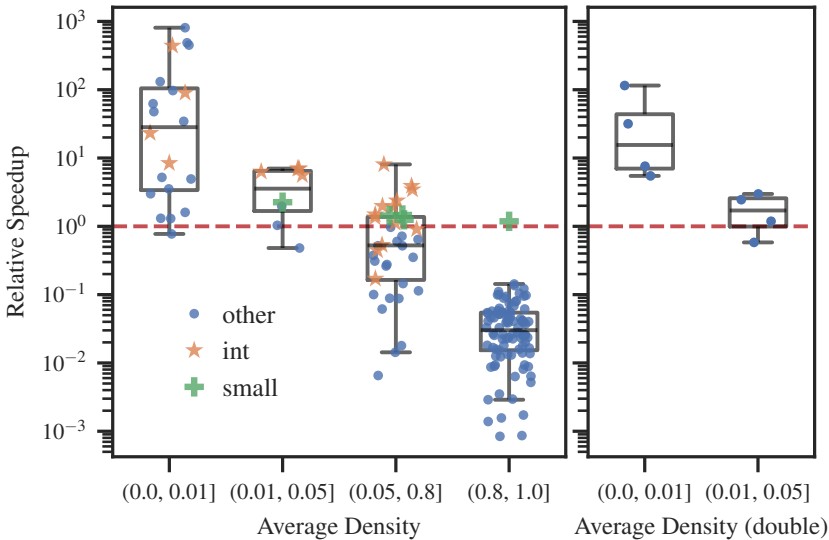

Figure 4: Relative speedup of the sparse over the dense tensor format on the benchmark dataset. The red line marks equal performance. The orange dots represent instances with an integer data type, and the green dots represent instances with fewer than $10^8$ flops. The right plot shows the relative performance on the sparse integer instances after conversion to the double data type.

**Outliers.** There are some outliers where sparse is faster even though the average density is high, primarily for two reasons. First, some of the instances, marked in orange, have very few flops. Therefore, the choice of tensor format has little impact. Our sparse implementation is written purely in C++, and thus has lower overhead during einsum expression evaluations, which leads to slightly better running times on these instances. Second, as shown in the benchmark paper [9], PyTorch can experience significant slowdowns on integer operations, whereas our sparse implementation does not have this problem, which results in a speedup for these instances. To validate the speedup for instances with densities below 5%, we converted all input tensors from integers to doubles and repeated the experiment. The speedup became smaller but persisted for sparse instances. Moreover, there are a few sparse outliers where PyTorch performs more efficiently, although their average density is below 5%.

### 4.4 Algorithmically exploiting dynamic sparsity

These observations show that sparsity in einsum is inherently dynamic: tensors that start sparse can create dense intermediates, tensors with few nonzeros can produce hyper-sparse intermediates later. There are no sparsity estimators tailored to this setting, and while existing cardinality estimators from the database literature are likely adaptable they are often unreliable and expensive on large expressions. Rather than relying on a costly upfront estimate, our hybrid approach detects sparsity during execution and adapts the representation on the fly. Below we first describe the hybrid algorithm in detail and then evaluate its effectiveness on the benchmark dataset.

**Hybrid algorithm.** The hybrid algorithm evaluates contractions in the dense format while the average density of the remaining (not-yet-contracted) tensors stays above a chosen threshold. Once the average density falls below that threshold, the algorithm switches to the sparse format. For example, in Figure 1 there are twelve remaining tensors on the left. After the first contraction the four center tensors are combined into a single intermediate tensor, leaving nine remaining tensors on

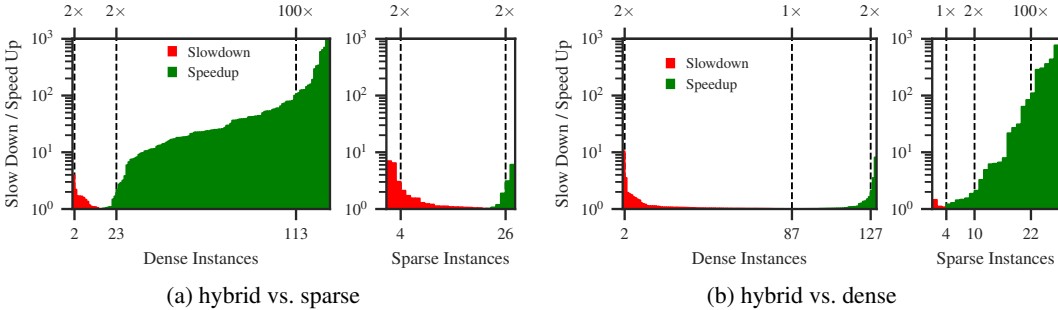

(a) hybrid vs. sparse          (b) hybrid vs. dense

Figure 5: Slowdown and speedup of hybrid algorithm on dense and sparse instances ($\text{avg}_{\tau_{\max}} < 0.05$).

the right. Because all other tensors are unchanged by this contraction, updating the average density requires only computing the density of the new intermediate tensor and updating the running totals.

For the experiments reported here we used a 5% threshold chosen empirically from the benchmark. As indicated by Figure 4, this threshold appears to be the point where a speedup is most likely. In general, the threshold depends mostly on the sparse implementation and should be chosen rather low, since this will only slightly delay the switch on instances that become hyper-sparse.

To minimize the overhead of calculating the density, we compute it only before expensive contractions. Moreover, once the average density exceeds 95%, we stop calculating it. In Figure 5, we compare the hybrid algorithm against the standard algorithm using either sparse or dense tensor formats.

**Hybrid vs. sparse.** The results of the comparison are shown in Figure 5a. The hybrid algorithm is significantly faster on the vast majority of dense instances. It is more than 100x faster on 17 instances, and even more than 1000x faster on three instances. Overall, we achieve a median speedup of 22x. We observe a small slowdown on small instances and on integer instances, which, as discussed earlier, are exceptional cases.

For sparse instances, the overhead of the hybrid over the sparse algorithm appears to be minimal. The median slowdown is approximately 1.08x. The hybrid algorithm takes twice as long for just four instances, primarily because it either does not switch to sparse or switches to sparse too late. However, switching too late only results in a slowdown due to performance differences observed on integer instances. There are also three instances that benefit from the hybrid algorithm because they are initially dense and thus benefit from the initial dense phase of the hybrid algorithm.

**Hybrid vs. dense.** The results of the comparison are shown in Figure 5b. In contrast to the comparison with the sparse algorithm, we cannot expect a speedup over the dense algorithm on dense instances, as additional operations are performed for the density test. However, in practice we observe a small speedup on the integer instances from the third group, where we incorrectly switch to the sparse format because of their mixed density characteristics that can be seen in Figure 3 (right). The largest slowdowns are experienced on the remaining instances of the third group, where we also incorrectly switch to the sparse format. On the other instances, the slowdown, and thus the overhead, on dense instances is indeed very small. The median slowdown is approximately 1.01x.

On sparse instances, the hybrid algorithm clearly outperforms the dense algorithm, reaching a speedup of nearly 1000x on a very sparse instance. The median speedup is approximately 6.2x. The hybrid algorithm is notably slower than the dense algorithm on only one sparse instance, because the dense implementation performs better even though the instance is sparse.

**Hybrid vs. virtual best.** Overall, the hybrid algorithm closely matches the runtime of the faster static backend (the virtual best between dense and sparse) and occasionally outperforms it. The density-estimation overhead is negligible in practice and provides clear gains on sparse instances. Measurable slowdowns occur mainly for mixed-density cases and integer-typed inputs. As shown in Figure 6, the hybrid algorithm is more than twice as slow as the virtual best for only 8 of 158 instances, and the median slowdown is about $1.02\times$ ($\approx 2\%$). Conversely, the hybrid method can beat both static choices: in our experiments it was substantially faster than the virtual best on 3 of the 28 sparse instances.

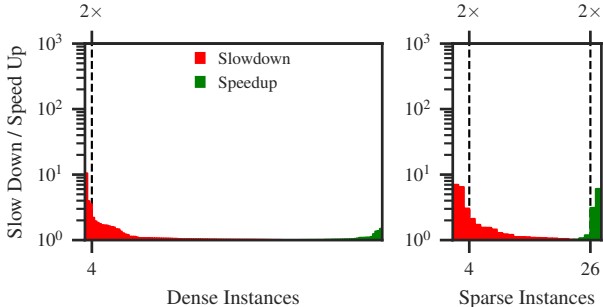

Figure 6: Slowdown and speedup of hybrid algorithm vs. the virtual best algorithm.

## 5 Limitations

While our hybrid approach to dynamically exploiting sparsity in einsum expressions demonstrates strong empirical performance across a range of benchmark instances, it also presents several limitations. First, the effectiveness of the hybrid strategy is sensitive to the choice of sparsity threshold used to trigger format switching. Although our 5% threshold performs well across the dataset, this value may not generalize across all domains or applications. A more adaptive or learned switching criterion could further improve performance but was not explored in this work.

Second, our current implementation only supports switching from dense to sparse formats, and not vice versa. As demonstrated by mixed-density instances, some tensors become sparse mid-evaluation but later become dense again. For such cases, the optimal strategy may involve switching back and forth between sparse and dense formats based on the evolving average density. Our hybrid approach does not support this bidirectional switching, which limits its flexibility. Moreover, some contractions might benefit from mixing sparse and dense tensors. Supporting mixed-format operations could further improve performance but is beyond the scope of this work.

Third, our experiments concentrate on CPU performance because sparse einsum support on accelerators (such as GPUs and TPUs) is currently too limited. Nevertheless, the hybrid algorithm is device-agnostic in design and can be adapted to other hardware. In fact, different devices could be used for the dense and sparse phases, depending on which device has better support for the respective format. However, we chose not to mix hardware in our experiments, as we consider the lack of sparse GPU einsum support a temporary limitation.

Finally, our method assumes a static contraction order, which may not be optimal when sparsity evolves dynamically. Integrating contraction order optimization that is aware of evolving sparsity could offer further improvements but is non-trivial and left for future work.

## 6 Conclusions

Einsum is a key component of modern machine learning frameworks and also has applications in classical areas of artificial intelligence, such as probabilistic models, probabilistic inference, and constraint satisfaction. The range of tensor expressions that einsum implementations must support is extensive. Sparsity in einsum has traditionally been handled statically, meaning a dense or sparse data structure is chosen before the evaluation begins. However, it is difficult to predict the most efficient data structure for a given einsum expression, especially when it involves many input tensors. Our experiments show that sparsity often arises dynamically in real-world einsum instances, with intermediate tensors becoming sparse during the evaluation, even if the input tensors are initially dense. To address this, we introduced a hybrid algorithm that dynamically switches the data structure from dense to sparse based on the sparsity of the remaining tensors. The runtime of density checks is negligible compared to the overall time required to evaluate the expression. Our hybrid algorithm is up to three orders of magnitude faster than using an inefficient data structure and performs comparably to or faster than the optimal static choice.

## Acknowledgments

This work was supported by the Carl Zeiss Foundation within the project Interactive Inference.

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
