# OpenReview forum: "Exploiting Dynamic Sparsity in Einsum"
_NeurIPS.cc/2025/Conference — NeurIPS 2025 poster_

### Official Review · Reviewer_DRa5 · 2025-06-24

**Clarity:** 3
**Significance:** 3
**Originality:** 2
**Rating:** 5
**Confidence:** 3

**Summary:**

The paper introduces the problem of choosing the correct data structure when evaluating einsum expressions on sparse data, particularly between tensors and coordinate arrays. They show on a family of grid formulae that the latter uses exponentially more flops than the former when contracting the representation of a grid. Then, experiments are performed to empirically support this.
Furthermore, they propose an hybrid algorithm that switches the representation on the fly, showing positive results.

**Questions:**

- Is it possible to apply the hybrid approach to the actual training of a neural network? For example to a simple CNN on MNIST to see the benefits regarding training time and resource use.
- Are there any related works that discuss the efficiency of structures on neural networks? It would be good to see if other approaches were considered before.

**Ethical Concerns:**

["NO or VERY MINOR ethics concerns only"]

**Final Justification:**

The paper offers a principled analysis of dynamic sparsity in einsum evaluation, combining theoretical insight with a practical hybrid implementation. The method is well-motivated, and the experimental results align with the theoretical claims. I appreciate the clarifications provided in the rebuttal, especially regarding backend generality and use cases in neuro-symbolic reasoning. I maintain my positive score and increase my confidence.

**Limitations:**

Yes, in Section 5.

**Paper Formatting Concerns:**

Nothing to report.

**Quality:**

3

**Strengths And Weaknesses:**

Strengths

- The paper addresses an important point about the efficiency of the structures we use during long routines of computation, which can lead to dramatic improvement in time and energy consumption.
- The empirical results are founded in a proved proposition and align with them.The benchmark of einsum instances seems broad and has been recently published.
- Well written and easy to read.

Weaknesses

- I find it strange that there is no related work. Although the approach seems completely novel, there must be research that speaks about transitions of structures with respect to some threshold. Adding some about this would improve to contextualize the originality of the approach and also to improve my confidence on the score.
- Some of the limitations could be addressed in the main article such as switching back to a dense format, which should be relatively easy to implement, or experiments on GPUs, which are essential for high performance computation in the context of AI.


Minor comments

- Figure 2. Grid Site Length (n) -> Size. Also, why is dense not present on the right subfigure? I think the subfigures should be better explained.
- Figure 3. Legend missing.

---

> ### Author Rebuttal · Authors · 2025-07-28
>
> Thank you for reviewing our paper and your questions. We appreciate the opportunity to address them.
>
> **Question 1:** *Is it possible to apply the hybrid approach to the actual training of a neural network? For example to a simple CNN on MNIST to see the benefits regarding training time and resource use.*
>
> The dynamic switching between dense and sparse data structures based on average density that we introduce here is potentially also useful for multi-tensor operations used in CNNs. However, the benefits of dynamic switching become more pronounced for *einsum* expressions that involve many tensors.
>
> **Question 2:** *Are there any related works that discuss the efficiency of structures on neural networks? It would be good to see if other approaches were considered before.*
>
> Dynamic sparsity is a known phenomenon in deep learning, see References  [17, 21, 24, 31, 33, 34, 37] in the paper. Traditionally, however, deep learning employs *einsum* expressions with only a few tensors. Only recently have large *einsum* expressions been used in machine learning areas such as probabilistic circuits, probabilistic neural circuits, and neuro-symbolic models that combine deep learning with logical constraints. Therefore, to the best of our knowledge, we are the first to study dynamic sparsity in large *einsum* expressions.

---

> ### Comment · Reviewer_DRa5 · 2025-08-01
>
> - R1: If I understood correctly looking at the other reviews, the method as it is cannot be used for machine learning on GPUs due to the nonexistence of sparse einsum implementations over these. Later you mention that einsum expressions have been used in the context of neurosymbolic computation. Please clarify where einsum expressions are particularly used to contextualise the contribution.
> - R2: To understand how the problem has become relevant, it would be good to see in more detail where and how einsum expressions are used.

---

> > ### Author Response · Authors · 2025-08-04
> >
> > **Answer to R1**
> >
> > It is correct that efficient generic sparse einsum is not yet available on GPUs. This is a limitation of current library support, not of our method. Our hybrid strategy is backend-agnostic and would apply equally well to GPUs or TPUs once sparse einsum becomes available. GPU-CPU hybrid execution is also feasible today. For example, switching to a sparse CPU implementation when tensors become sparse on the GPU. Our method supports such transitions naturally. However, we chose not to mix hardware in our experiments, as we consider the lack of sparse GPU einsum support a temporary limitation. We will clarify in the paper that the approach is portable across hardware backends.
> >
> > For an application of einsum in neuro-symbolic AI: Calanzone et. al [10] turn complex reasoning tasks into logical constraints that can be incorporated via neuro-symbolic reasoning into an LLM. The LLM is fine tuned by "a principled objective: maximising the probability of a constraint to hold, which is known as weighted model counting." (Large scale) weighted model counting is a fundamental task that can be directly mapped to einsum. The benchmark dataset contains 14 weighted model counting instances, seven of which become sparse. Moreover, our theoretical example in Section 3 is a family of model counting problems.
> >
> > [10] Diego Calanzone, Stefano Teso, and Antonio Vergari. Logically Consistent Language Models via Neuro-Symbolic Integration. In ICLR, 2025.
> >
> > **Answer to R2**
> >
> > The core feature of einsum is breaking down a large (output) tensor into many smaller (interacting) input tensors. Therefore, einsum can be used to represent logical structure, probabilistic dependencies as in probabilistic circuits, or quantum interactions in quantum machine learning. Dynamic sparsity often arises during execution in these settings, even when input tensors are dense. It is used also but only to some extent, in deep learning architectures, though in the latter case, large einsum expressions are still less common.
> >
> > As noted in our response to R1, weighted model counting is one example where large (sparse) einsum expressions arise. Another example is the simulation of quantum circuits, where the initial state and the gates are represented as tensors, leading to one large einsum expression. The output tensor is the final state of the quantum system. Using einsum (tensor networks) enables state-of-the-art quantum simulation [A1, A2]. As access to quantum computers is limited, simulating quantum circuits is essential for quantum machine learning [7]. The benchmark contains 32 quantum circuit instances, of which 3 become hypersparse, with an average density of less than 0.0001%.
> >
> > An area in deep learning that fits einsum well are LoRAs (low-rank adaptation) [A3], which inject trainable rank decomposition matrices into the layers of the transformer architecture. The rank decomposition matrices can be represented by tensor networks and thus by einsum. So far mostly small factorizations have been used. However, a recent survey [A4] discusses the relationship between low rank tensor decompositions and probabilistic circuits which traditionally have been much deeper. It suggests a "Lego blocks" approach of stacking multiple low-rank decompositions to form larger decompositions and thus einsum expressions.
> >
> >
> > [A1] Huang, C., Zhang, F., Newman, M. et al. Efficient parallelization of tensor network contraction for simulating quantum computation. Nat Comput Sci 1, 578–587 (2021).
> > [A2] Morvan, A., Villalonga, B., Mi, X. et al. Phase transitions in random circuit sampling. Nature 634, 328–333 (2024).
> > [A3] Edward J. Hu, Yelong Shen, Phillip Wallis, Zeyuan Allen-Zhu, Yuanzhi Li, Shean Wang, Lu Wang, Weizhu Chen. LoRA: Low-Rank Adaptation of Large Language Models. In ICLR 2022
> > [A4] Loconte, L., Mari, A., Gala, G., Peharz, R., de Campos, C., Quaeghebeur, E., Vessio, G., & Vergari, A. (2025). What is the Relationship between Tensor Factorizations and Circuits (and How Can We Exploit it)? Transactions on Machine Learning Research, 2025(02).

---

> > > ### Comment · Reviewer_DRa5 · 2025-08-06
> > >
> > > I thank the authors for their detailed rebuttal.
> > > I will keep my positive score and improve my confidence.

---

### Official Review · Reviewer_hrqu · 2025-06-29

**Clarity:** 3
**Significance:** 2
**Originality:** 3
**Rating:** 5
**Confidence:** 3

**Summary:**

This paper evaluates the benefits of sparse tensor storage for einsum computations.
On the theoretical side, the authors prove that for a well-chosen family of instances, dense storage can require exponentially more FLOPs for tensor contraction than sparse storage.
On the practical side, they perform an analysis of sparsity evolution as contraction progresses, for a recently-introduced benchmark ([https://benchmark.einsum.org/](https://benchmark.einsum.org/)). This analysis shows that sparsity can sometimes appear halfway through the process, even when the input tensors are dense. Therefore, a new storage strategy is proposed, which switches from dense to sparse when the average density of the tensor sequence falls below a pre-specified threshold. Numerical experiments demonstrate that this strategy can lead to efficiency gains.

**Questions:**

- To justify a dynamic approach, the authors state that “in einsum expressions involving a large number of tensors, it is not known in advance whether intermediate tensors will become sparse.” Is that obvious? Are there heuristics that could guide a static storage choice?
- Describe the contraction orders used in the experiments and their impact on the storage and FLOPs requirements.
- Explain more precisely how the “not-yet contracted tensors” are defined and how their average density is calculated.
- Add a comparison between the hybrid algorithm and the best of both dense and sparse algorithms, in order to back the claim from the abstract.
- Defend your choice of sesum as a sparse einsum library, as opposed to the Python sparse library you mention in the introduction.

---

Satisfactory answers to most of these questions can lead me to increase my rating to accept.

**Ethical Concerns:**

["NO or VERY MINOR ethics concerns only"]

**Final Justification:**

Raising my score to 5 following a convincing rebuttal, to make it clear that this paper deserves acceptance despite the concerns raised about GPU compatibility.

**Limitations:**

Yes.

**Paper Formatting Concerns:**

None.

**Quality:**

2

**Strengths And Weaknesses:**

## Strengths

### Quality

- The separation theorem on model counting for logical formulas is a strong theoretical motivation for sparse storage.
- The measurement of dynamic sparsity in the Einsum Benchmark provides empirical justification for an adaptive approach.

### Clarity

- The paper is well-written.
- The experiments are reproducible.

### Significance

- Tensor contraction is a thriving topic in probabilistic inference, and this study opens up several promising research avenues, especially in Section 5.

### Originality

- The focus on CPU operations is unexpected in machine learning but justified for handling dynamic sparsity.
- The adaptive storage policy is rather naive but it is possible that no one has benchmarked it before.

## Weaknesses

### Quality

- The contraction order used in the experiments is never discussed in the main paper. In the appendix, we only read that “The benchmark includes one contraction order optimized for minimal intermediate size and one for minimal floating-point operations (flops).”
- The abstract claims that “In our experiments on established benchmark einsum expressions, the hybrid solution is consistently competitive with or outperforms the better of the two static representations.”, but the hybrid algorithm is only compared to its purely dense and sparse counterparts *separately* (Figure 5). It should also be compared to the best of both for each instance.

### Clarity

- It is not clear to me how the hybrid algorithm defines “the average density of *the not-yet contracted tensors*”. After all, a tensor could be involved at several steps of the contraction order (which is defined on axes, if I understand correctly). To put it differently, contractions after step 1 can be defined on intermediate tensors which do not yet exist at the beginning of the process.

### Significance

- In the experiment code, a sparse tensor library called sesum (which I wasn’t able to find elsewhere) is apparently used as a binary wheel, which makes the experiments less reusable and generalizable.

---

> ### Author Rebuttal · Authors · 2025-07-27
>
> Thank you for reviewing our paper and your questions. We appreciate the opportunity to address them.
>
> **Question 1:** *To justify a dynamic approach, the authors state that “in einsum expressions involving a large number of tensors, it is not known in advance whether intermediate tensors will become sparse.” Is that obvious? Are there heuristics that could guide a static storage choice?*
>
> Unfortunately, this is not obvious. Sparsity in *einsum* is a truly dynamic property, and all the following are possible: 1. Sparse input tensors can lead to dense intermediate tensors. 2. Dense input tensors with only a few non-zero entries can lead to intermediate and result tensors that are hypersparse. 3. Sparse intermediate tensors can lead to dense tensors later in the contraction order.
>
> Sparsity estimation techniques such as cardinality estimation, which are used in the context of databases, tend to be expensive and unreliable for large expressions. To the best of our knowledge, there are no sparsity estimation techniques specifically for *einsum*. Therefore, sparsity estimation for einsum is a promising avenue for future research. We will address this in the paper.
>
> **Question 2:** *Describe the contraction orders used in the experiments and their impact on the storage and FLOPs requirements.*
>
> We use the contraction orders that are provided with the *einsum* benchmark, Reference [9] in the paper. We summarize key metrics of the contraction paths in Table 2 of the supplement. Regarding impact, dynamic sparsity is contraction order dependent. From our observations, flop-optimized contraction orders tend to result in sparser tensors than size-optimized orders. Therefore, we used flop optimized paths, except for problems that become too large to fit into RAM in the dense evaluation case.
>
> **Question 3:** *Explain more precisely how the “not-yet contracted tensors” are defined and how their average density is calculated.*
>
> Any tensor, input or intermediary, is contracted only once, because a contraction replaces the participating tensors by a new intermediary tensor. This can be explained using the example that is shown in Figure 1 in the paper. Before the evaluation of the *einsum* expression starts, twelve tensors (black circles) that are shown in Figure 1(left) are all not-yet contracted. Then, after the first contraction, shown in Figure 1(right), the four tensors at the center are contracted into an intermediate tensor resulting in nine tensors: eight original tensors and one new intermediate tensor. These nine tensors are not-yet contracted after the first contraction. Therefore, the density of the not-yet contracted tensors before the first contraction is the average density of the input tensors and the average density of the not-yet contracted tensors after the first contraction is the average density of the remaining nine tensors. Thus, the only new density that has to be computed after the first contraction is the density of the new intermediate tensor. The average density can then be updated accordingly. We will clarify this in the paper.
>
> **Question 4:** *Add a comparison between the hybrid algorithm and the best of both dense and sparse algorithms, in order to back the claim from the abstract.*
>
> Besides the outlier instances, the comparison of hybrid vs the best of sparse and dense boils down to the plots of sparse instances using the sparse backend and dense instances using the dense backend. We will include a figure that shows this explicitly in supplemental material. Overall, the hybrid algorithm is only more than twice as slow as the best in eight out of 158 instances. In the median, we see a slowdown of roughly 1.02x, so about 2 percent slower, than the best of both. Notably, the hybrid approach can also be significantly faster than the best of sparse and dense. In our experiments, that happened on three out of 28 sparse benchmark problems. We will add a figure to the supplement to highlight this.
>
> **Question 5:** *Defend your choice of sesum as a sparse einsum library, as opposed to the Python sparse library you mention in the introduction.*
>
> In the Python sparse library, each index is encoded in a separate integer. It turns out that this slows the evaluation of *einsum* expressions with tensors that have many indices. In the *sesum* implementation of *einsum*, the indices of a tensor are encoded in a single number, which leads to a smaller memory footprint and faster execution.
>
> Even on sparse instances with 1% non-zeros, such as *mc_2022_029* or *mc_rw_blasted_case1_b14_even3* from the benchmark, the Python sparse library can be significantly slower than PyTorch's dense *einsum* implementation.
>
> For the reproducibility of our experiments, *sesum* is included in the project repository. Running `pip install sesum-0.3.8-py3-none-any.whl` or `uv sync` will make it available on any Linux or macOS machine.

---

> ### Comment · Reviewer_hrqu · 2025-08-04
>
> Thank you for your answers. A few follow-ups:
>
> - More details on the contraction orders used should be added to the appendix, especially because they influence sparsity.
> - Where does `sesum` come from? Did you author it? A quick Google search did not allow me to find it elsewhere.
>
> After reading the other reviews and rebuttals, I conclude that the authors did a good job of distinguishing between a current software limitation (the lack of GPU implementations for sparse `einsum`) and a fundamental property of tensor networks (dynamic sparsity and its impact), while studying the latter in detail. Besides, not all machine learning has to happen on a GPU. I will raise my score to 5 to make it clear that I find this paper self-sufficient and ready to be accepted.

---

> > ### Author Response · Authors · 2025-08-04
> >
> > Dear reviewer, thank you for the positive feedback and for raising your score.
> >
> > - *More details on the contraction orders used should be added to the appendix, especially because they influence sparsity.*
> >
> > Thank you, that is good advice. We will do so.
> >
> > - *Where does `sesum` come from? Did you author it? A quick Google search did not allow me to find it elsewhere.*
> >
> > Yes, we have authored sesum and will release it to the public soon.

---

### Official Review · Reviewer_FHEA · 2025-07-03

**Clarity:** 3
**Significance:** 2
**Originality:** 2
**Rating:** 4
**Confidence:** 4

**Summary:**

This paper proposed a hybrid dense-sparse Einsum for computing large Eisnum expressions. The authors show that there exist non-trivial examples of Einsum programs, in the form of ${\rm Grid}_n$, where the popular dense implementation is much slower than a coordinate-list-based sparse implementation. Through experiments, the authors show that their proposed hybrid dense-sparse Einsum is much performant that the baseline dense implementation.

**Questions:**

- It'd be better to spell out, in pseudocode, the algorithms for contraction for both sparse and dense tensors.
- Specify what the ${\rm Grid}$ model is used for. E.g., for modeling magnetic crystal structures? Given more context here would aid understanding of the setting of the method.
- L147: ${\rm Grid}_n$: Provide a large illustration / figure of this tensor network.
- L148: Please add the definition $[n] = \\{i \in \mathbb{N} | 0 \le i < n\\}$.

**Ethical Concerns:**

["NO or VERY MINOR ethics concerns only"]

**Final Justification:**

I've read the comment and had a better understanding of the main contributions of the paper. I'm happy to increase the scores of the paper.

**Limitations:**

yes

**Quality:**

2

**Strengths And Weaknesses:**

Strengths:
- Simple method that keeps track of the proportion of non-zero elements in tensors; and switch to sparse when below a threshold. This method is simple and performs well.
- Detailed exposition on the ${\rm Grid}_n$ tensor network.

Weaknesses:
- The background is too long in the paper, where the proposed solution appeared at Section 4.4, page 8. The hybrid algorithm is very simple (if non-zero entries >5%, dense). The substance of this paper does not seem to merit a long NeurIPS submission.
- The proposed method works for ${\rm Grid}$-type tensor networks, but actual tensor Einsum programs take many different forms (in deep learning; belief propagation; quantum computing; etc). The performance might not generalize to other forms of tensor networks. A good benchmark for this is the Einsum benchmark (https://benchmark.einsum.org/): https://proceedings.neurips.cc/paper_files/paper/2024/file/b1bbfdb9197bfc819a52c34dce493f85-Paper-Datasets_and_Benchmarks_Track.pdf, which I believe that the authors should do experiments on.
- Tensor operations are now predominantly performed in a parallel way on specialized hardwares such as GPUs or TPUs. Analysis should be done over these heterogeneous hardware.

---

> ### Author Rebuttal · Authors · 2025-07-28
>
> Thank you for reviewing our paper. Unfortunately, there are some inaccuracies in your review that we address below. We hope you take this into account and update your score accordingly.
>
> **Weakness 1:** *The proposed method works for Grid-type tensor networks, but actual tensor Einsum programs take many different forms (in deep learning; belief propagation; quantum computing; etc). The performance might not generalize to other forms of tensor networks. A good benchmark for this is the Einsum benchmark, which I believe that the authors should do experiments on.*
>
> As we write in the second sentence of the experimental section (Section 4), our experiments were performed on this benchmark, which covers several application domains.
>
> **Weakness 2:** *The background is too long in the paper, where the proposed solution appeared at Section 4.4, page 8. The hybrid algorithm is very simple (if non-zero entries >5%, dense). The substance of this paper does not seem to merit a long NeurIPS submission.*
>
> We do not consider the hybrid algorithm the main contribution of our paper, but rather the theoretical and practical analysis of dynamic sparsity in *einsum*. Therefore, we do not consider Section 3 as background material but as a contribution. The Grid example in Section 3 shows that there exist *einsum* expressions where the two data structures, dense and sparse, are exponentially separated, that is, the dense data structure dynamically becomes exponentially larger than the sparse one, whereas the overhead of sparse over dense is always polynomially bounded. Then, in Section 4, we show on the *einsum* benchmark that a separation can also be observed on practical *einsum* problems. Moreover, we show that dynamic sparsity can be exploited to speed up the evaluation of *einsum* expressions.
>
> **Weakness 3:** *Tensor operations are now predominantly performed in a parallel way on specialized hardwares such as GPUs or TPUs. Analysis should be done over these heterogeneous hardware.*
>
> We want to distinguish between the phenomenon of dynamic sparsity in *einsum* and implementations that exploit this phenomenon. Dynamic sparsity arises independently of the compute backend, that is, CPU, GPU, or TPU. The proposed method would work on TPUs and GPUs as well, but so far there are no established sparse *einsum* implementations for GPUs and TPUs.
>
> **Question/Comment 1:** *It'd be better to spell out, in pseudocode, the algorithms for contraction for both sparse and dense tensors.*
>
> Contraction orders break down an *einsum* expression into a sequence of binary contractions, that is, contractions of two tensors. Implementationd of binary contraction algorithms in either case, dense and sparse, are highly advanced and still an active area of research. They are not in the focus of our paper.
>
> **Question/Comment 2:**  *Specify what the model is used for. E.g., for modeling magnetic crystal structures? Given more context here would aid understanding of the setting of the method.*
>
> The Grid model is an instance of a #SAT (model counting) problem. Model counting has applications, among others, in probabilistic inference. For instance, the inference engine of the probabilistic programming language DICE uses model counting in its backend. However, here, the Grid model only serves as an example of a tensor contraction problem that exhibits dynamic sparsity. It serves as a formal proof that the difference between sparse and dense representations can be exponential.
>
> **Question/Comment 3:** *L147: Provide a large illustration / figure of this tensor network*
>
> The Grid$_3$ model is illustrated in Figure 1.
>
> **Question/Comment 4:** *L148: Please add the definition of $[n]$.*
>
> We will do so.

---

> > ### Comment · Reviewer_FHEA · 2025-08-08
> >
> > Thanks for the clarification for what contributes the main ideas of the paper. I will modify the scores.

---

### Official Review · Reviewer_pjM9 · 2025-07-03

**Clarity:** 3
**Significance:** 3
**Originality:** 3
**Rating:** 4
**Confidence:** 2

**Summary:**

This paper talked about the limitations and constrains in tensor computing, where they modify tensor hypernetworks to speed up the computing process. The proposed method showed a great performance increase across tensors with different densities.

**Questions:**

Please see previous section.

**Ethical Concerns:**

["NO or VERY MINOR ethics concerns only"]

**Final Justification:**

This paper brings efficient solutions in computing. I will keep the score as 4.

**Limitations:**

Yes

**Quality:**

3

**Strengths And Weaknesses:**

Strengths:
1. This paper discussed the constrains and limitations in (sparse) tensor computations.
2. They modified the tensor hypernetworks to speed up the computation. Such modification is suitable for tensors with varied densities.
3. They provide a small C++ module plugs into PyTorch/opt_einsum for CPU capability.

Weaknesses:
1. One limitation I noticed and the author pointed out in the limitation section is how would this method work on GPU. Most tensor computation is utilizing GPU. If the proposed method can only work on CPU. It could greatly limit the applicability of the proposed method.
2. Can the proposed method support sparse to dense fallback?
3. How sufficient is the 5% threshold. How sensitive are results to this value?

---

> ### Author Rebuttal · Authors · 2025-07-27
>
> Thank you for reviewing our paper. We appreciate the opportunity to address the weaknesses that you have mentioned.
>
> **Weakness 1:** *One limitation I noticed and the author pointed out in the limitation section is how would this method work on GPU. Most tensor computation is utilizing GPU. If the proposed method can only work on CPU. It could greatly limit the applicability of the proposed method.*
>
> We want to distinguish between the phenomenon of dynamic sparsity in *einsum* and the implementations that exploit it. Dynamic sparsity arises independently of the compute backend, that is, CPU or GPU. The proposed method would work on GPUs as well, but so far there are no established sparse *einsum* implementations for GPUs.
>
> **Weakness 2:** *Can the proposed method support sparse to dense fallback?*
>
> Yes, this is possible. In this work, we kept the setup simple and focused on analyzing the phenomenon of dynamic sparsity in *einsum*. Nevertheless, our motivation is practical, and we will explore all options as we further engineer an efficient sparse *einsum* implementation that exploits dynamic sparsity.
>
> **Weakness 3:** *How sufficient is the 5% threshold. How sensitive are results to this value?*
>
> In general, the threshold should not be chosen too large, because, as can be seen in Figure 3, sparsity can first decrease and then increase again, which can result in large performance overheads depending on the performance of the sparse *einsum* implementation. Switching later than necessary because of a threshold that is too small incurs only small overheads as long as the switch happens before the large, sparse contractions that typically dominate the end of the computations. On the benchmark, the optimal value is around 5%. Setting it as low as 1% would affect fewer than ten instances, all with minor sparse speed-ups.

---

> > ### Comment · Reviewer_pjM9 · 2025-08-05
> >
> > I appreciate the response from the author.
> > In general, I think this is well written paper. I will keep my score after reviewing the response.
> > The reason being:
> > 1. I do not think the implementation of the dynamic sparsity in einsum could be independent from the backend. As how you construct the logic circuits may work on CPU but not necessarily on GPU.
> > 2. I understand the sparsity would show fluctuation during the calculation. But without further experiments, the 5% threshold is not entirely justified.

---

> > > ### Author Response · Authors · 2025-08-06
> > >
> > > We appreciate your positive assessment of our paper. Regarding backend dependence, we would like to clarify that dynamic sparsity is a mathematical property of the einsum expression and not of the hardware. The same switching logic applies whether the backend is CPU, GPU or TPU, with only the underlying sparse implementation changing and the switching threshold potentially needing adaptation. For our setting and the einsum benchmark, 5% proved especially suitable. That said, you are correct that the threshold is not fully justified and will depend on the specific dense and sparse einsum backends used. We will incorporate this clarification into the paper to make it clear that sparsity is problem inherent rather than backend driven, and that the 5% threshold is not hardwired but expected to be adapted for different hardware and software environments.

---

### Decision · Program_Chairs · 2025-09-17

**Decision:**

Accept (poster)

**Comment:**

The paper investigates dynamic sparsity in einsum computations, demonstrating a potential exponential gap between dense and sparse representations, and proposes a simple hybrid algorithm that switches between the two representations when density falls below a threshold. Experiments demonstrate that the hybrid solution is consistently competitive with, or outperforms, the better of the two static approaches.

The work offers a novel and well-founded treatment of dynamic sparsity as a hardware-agnostic property, combining theory with reproducible experiments. The authors convincingly addressed critiques on scope, writing, and related work during the rebuttal. Overall, the contribution is technically sound, relevant across domains, and is therefore recommended for acceptance.